# Creativity and Personality Traits as Foreign Language Acquisition Predictors in University Linguistics Students

**DOI:** 10.3390/bs10010035

**Published:** 2020-01-15

**Authors:** Irina A. Novikova, Nadezhda S. Berisha, Alexey L. Novikov, Dmitriy A. Shlyakhta

**Affiliations:** 1Social and Differential Psychology Department, Peoples’ Friendship University of Russia (RUDN University), 6 Miklukho-Maklaya Str., 117198 Moscow, Russia; 2Department of Political and Legal Disciplines and Social Communications, The Russian Presidential Academy of National Economy and Public Administration, 82/1 Vernadskogo Av., 119571 Moscow, Russia

**Keywords:** creativity, personality traits, foreign (second) language acquisition, ATTA, Five-Factor Model, linguistics students, English as a foreign (second) language, level of FL proficiency

## Abstract

Foreign (second) language (FL/SL) proficiency is one of the most important competencies for a modern person, and is necessary for both professional and personal fulfillment. The purpose of this study is to consider and compare personality traits and creativity as predictors of success in foreign language acquisition (FLA). The sample includes 128 (105 female and 23 male) first- and second-year university linguistics students. Creativity is measured by the Abbreviated Torrance Test for Adults (ATTA). The FFM personality traits are measured by the Russian NEO Five-Factor Inventory adaptation by S. Biryukov and M. Bodunov. To assess the level of FL proficiency, we used a traditional academic achievement indicator (the semester’s final grades in English), as well as the English teachers’ assessment of the level of language proficiency of their students according to the “Foreign Language Proficiency Scale” (10 indicators and total score). Descriptive statistics methods and a multiple regression analysis were used to process the data in the R software environment, version 3.5.2. The findings of our research showed that creativity indicators have a stronger but contradictory impact on the level of FL proficiency compared to personality traits. We suggest that teachers, most likely, lack knowledge on the manifestations of student creativity in the process of FL learning.

## 1. Introduction

The search for predictors of success in foreign (second) language acquisition (FLA/SLA) is an important interdisciplinary problem that has stimulated educators, teachers, psychologists, linguists, and specialists in intercultural communication to find a solution [1,2,3,4,5,6,7]. The development of this problem is of particular practical importance in the context of a learner-centered approach [4,5]. From the standpoint of this approach, considering the individual characteristics of a student will allow him/her to become a more independent, self-governed learner, which will contribute to FLA/SLA success [4,5]. Numerous studies show that these individual psychological characteristics include, above all, motivation, language aptitude, cognitive processes (features of perception, memory, intelligence, creativity, etc.), personality traits, and communication strategies or styles of FL learning [1,2,4,5,6,7]. In this article, we consider, on the one hand, personality traits, which are traditionally studied as the most important factors for educational achievements in general, and as predictors of FLA/SLA in particular, and on the other hand, creativity, which, in our opinion, is less commonly studied in detail as a FLA/SLA predictor, especially in Russian psychology. Although the relationship between different personality traits and creativity is well studied [8,9], these variables have rarely been studied together as predictors of academic achievement. The research by Chaoyun Liang and Wei-Sheng Lin studied the combined effects of creativity, imagination, and personality traits on the academic performance of design students and revealed that flexibility has the strongest indirect effect on academic performance, followed by conscientiousness, initiating imagination, transforming imagination, and extroversion [10]. Since similar studies on the combined effects of creativity and personality traits on FLA/SLA are unknown to us, we have carried out our present research.

In Western psychology, personality traits as FLA/SLA predictors are often studied using the Five-Factor Model (FFM) developed by Robert R. McCrae and Paul T. Costa [11,12,13]. The FFM includes such factors as neuroticism, extraversion, openness to experience, agreeableness, and conscientiousness, which are described in the literature [11,12,13,14]. Numerous studies show that openness to experience and especially conscientiousness are significant predictors of academic achievements for most studied subjects, including FL/SL [15,16,17]. Recent Russian research based on the FFM has shown that conscientiousness impacts the academic achievements of Russian university and grade-school students [18,19,20,21]. However, a study of Russian linguistics students’ academic achievements in different subjects showed that none of the FFM traits are associated with a student’s final grades in English (as the “first” and mandatory FL), while conscientiousness, extraversion, and neuroticism are correlated with the final grades for their “second” elective FL [18]. In our previous study, we found that openness is most closely correlated with academic achievements in English as an FL among Russian linguistics students [21]. In Russian psychology, a large number of FLA studies have been carried out using the System–Functional Model of personality traits developed by Krupnov [22]. Researchers consider sociability, curiosity, persistence, responsibility, and initiative as the FLA factors for different university students [19,20,21,23,24,25]. In particular, it was revealed that persistence is more closely related to success in FLA than curiosity [19,20], and initiative has a contradictory correlation with FLA assessments in linguistics and non-linguistics university students [21,23]. These findings are consistent with those of other studies, which showed that students’ academic achievements are more strongly associated with conscientiousness than openness [15,16].

Creativity is often studied as a predictor of academic achievement [26,27,28,29,30]. However, some authors believe that creativity has been under-researched and somewhat ignored in the field of second or foreign language learning [31,32]. Creativity is complex in nature [33], and one of the complexities in defining the concept of creativity is the existence of different relevant notions of creativity, such as creative performances or products, creative people, creative situations, creative processes, and creative potential [31]. Many scholars and practitioners believe that one of the necessary changes in modern education should be the implementation and fostering of creativity in curricula, although not all teachers and students are ready for such changes [34]. In the context of FLA/SLA problems, a number of researchers have engaged in unpacking the relationship between bilingualism and creativity, and have found significant associations between them. On the one hand, bilingualism enhances cognitive functions such as planning, cognitive flexibility, and working memory; on the other hand, creativity is heavily dependent on the strength and power of these functions [35]. Kharkhurin proposed a Multilingual Creative Cognition Paradigm and considered bilingualism and multilingualism as factors in the development of human creativity [36,37]. Call et al. also reported on the significance of creativity in learning a SL/FL and overall language use [38]. Several researchers showed a significant relationship between creativity and language proficiency, and also determined that students with a high level of creativity had higher foreign language achievement scores [32,39,40]. However, Albert and Kormos revealed a contradictory relationship between different aspects of creativity and a learner’s performance in oral narrative tasks. While creative fluency proved to be positively correlated with better performance on the part of learners, originality negatively affected their performance in oral narrative tasks [41].

In our previous studies, we found only a few correlations between creativity and the level of FL proficiency among linguistics students [42,43]. In particular, we did not find any significant correlations between verbal creativity and FL proficiency assessments, or between all studied creativity indicators and final grades in English [42]. Also, we have determined that there are differences only in the flexibility among students with different levels of FL proficiency: Students who are more fluent in English show greater flexibility in solving creative tasks [43].

We assume that an explanation of these results may be connected to the following aspects: (1) The complex characteristics of the relationships between creativity and FL proficiency indicators, which may be mediated by other factors, including personality traits; (2) the ambiguous assessment of students’ creativity in FL learning by teachers, which corresponds to the assumptions by Shaheen [34]; and (3) the differences in the educational and evaluation systems used by different countries, such as the inconsistencies in the results from studying of personality traits as academic achievement predictors in Russian samples [18,19,20,21,23,24].

In summary, personality traits and creativity have been studied in detail separately as FLA/SLA predictors. Findings about the impact of personality traits are more consistent, while creativity impacts showed contradicting results. We believe that a joint review of these variables’ impacts on FL proficiency will help to both explain these contradictions and contribute to the development of practical recommendations for FL teachers in the context of a learner-centered approach.

Thus, the purpose of the present study is to consider and compare personality traits and creativity as predictors of FLA success in linguistics students. We hypothesize that among the various personality traits, conscientiousness and openness will have a positive impact on FL proficiency assessments, while different creativity indicators will have a more controversial impact. 

## 2. Methods

### 2.1. Participants

A total of 128 (105 female and 23 male) respondents aged 17 to 22 years (M = 18.67, SD = 0.98) took part in the research. All the respondents were first- and second-year bachelor students of the Department of “Linguistics” at the Institute of Foreign Languages (Peoples’ Friendship University of Russia). This curriculum includes both the study of languages and linguistic disciplines, as well as other subjects. The English Language is a basic subject, both in relevance and in the number of hours devoted to this subject. All students study English as their major FL. In this article, we use the term ‘linguistics student’ according to the name of the students’ department.

All participants engaged in psychodiagnostic tests (see Section 2.2) in accordance with the instructions in the presence of the experimenter after English classes. They were told that participation would be free and voluntary. 

In addition, 12 English teachers (100% female) participated in the study as experts. Each teacher assessed ten FL proficiency aspects among their students using the proposed scale (see Section 2.2).

All subjects gave their informed consent for inclusion before they participated in the study. The study was conducted in accordance with the APA Ethical Standards and the Code of Ethics of the RPS (Russian Psychological Society), and the protocol was approved by the Ethics Committee of RUDN University (# 050422-0-012).

### 2.2. Techniques

To measure the FFM traits, we used the Five-Factor Inventory (FFI), which is the Russian version of NEO-FFI adapted by S. Biryukov and M. Bodunov [44]. The FFI consists of 60 direct and inverted items, to which the subject expresses their degree of consent using a 5-point Likert scale (from “strongly disagree” to “strongly agree”). The resulting values for each of the five personality factors (neuroticism, extraversion, openness, conscientiousness, and agreeableness) range from 12 to 60 points [44]. This version of the questionnaire has been well tested on different Russian samples [19,21,45].

Creativity was measured using the Abbreviated Torrance Test for Adults (ATTA). The ATTA is designed to be a shortened version of the acclaimed Torrance^®^ Tests of Creative Thinking (TTCT), created specifically for use with adults [46]. The ATTA consists of three open-ended activities. The first test calls for verbal responses. This activity gives the researcher a chance to see how good the student is at creating new ideas and solving problems. After completing the task, three measures (fluency, originality, and the amount of special creativity indicators for verbal responses) are identified, summed, and interpreted as Verbal Creativity. The second and the third tests call for figural responses. Task #2 presents two incomplete figures and asks respondents to draw pictures of these figures, to attempt to make the pictures as unusual as possible, and to name these pictures. Then the four measurements (fluency, originality, elaboration, and the amount of the special creativity indicators for figural responses) are identified and summed (Figural Creativity #1). Task #3 presents 9 incomplete figures and asks respondents to draw pictures using these figures and to attempt to make the pictures as unusual as possible. The respondents are asked to make as many objects as he or she can and to remember to name the pictures. Then, five measurements (fluency, originality, elaboration, flexibility, and the amount of special creativity indicators for figural responses) are identified and added up (Figural Creativity #2). The Creativity Index (CI) is the sum of all listed measures [29]. For further calculations, we used the sum for each activity indicator separately (i.e., Verbal Creativity, Figural Creativity #1, and Figural Creativity #2) and the total indicators of Fluency, Originality, Elaboration, Flexibility, and CI.

To assess the level of FL proficiency, we used two procedures: (1) The academic achievement indicator (the semester’s final grades in English, which range from 0 to 100); and (2) the English teachers’ assessment of the level of language proficiency among their students, according to the “Foreign Language Proficiency (FLP) Scale” developed by us [21]. The FLP Scale includes 10 assessments. Four of these criteria (listening (audition), reading, writing, and speaking) correspond to the International English Language Testing System (IELTS) assessment scale [47]. Three more criteria (vocabulary, phonetics, and communication) are traditionally taken into account when assessing the level of FL proficiency in the Russian educational system (Russian students usually face difficulties while studying phonetics and practicing communication). According to the purpose of the study, which is meant to compare the impact of personality traits and creativity on FLA, we added the criteria of initiative, diligence, and creativity in learning English. All 10 indicators are presented in detail in Table 1. English teachers assessed their students using a quantitative 5-point scale (from 1 to 5) for each of the 10 indicators. The sum of all indicators is the Total Score (from 10 to 50). Psychometric testing using Cronbach’ *α* and McDonald’s *ω* coefficients showed that the proposed FLP Scale has a high degree of internal consistency for all points (*ω_h_* = 0.90, *α* = 0.64) [48,49]. At the same time, all parameters of the teachers’ evaluations had positive correlations with the semester’s final grades for English (*r* = 0.40–0.73, *p* ≤ 0.001). These data indicate the internal and external validity of the proposed FLP Scale [21]. 

### 2.3. Statistical Analysis

The descriptive statistics methods, Cronbach’ *α* and McDonald’s *ω* coefficients, Fisher’s *F*-test, and multiple regression analysis were used for the statistical analysis in the R software environment for statistical computing and graphics, version 3.5.2 [50]. We also used a regression analysis via the method of a “backward” or stepwise search. Independent variables included the FFM personality traits (FFI scales) and 8 creativity indicators measured by ATTA (Verbal Creativity, Figural Creativity #1, Figural Creativity #2, Fluency, Originality, Elaboration, Flexibility, and CI). Dependent variables included the level of FL proficiency assessments (10 indicators and Total Score according to the FLP Scale) and final grades in English. In the first step, full regression models with all possible predictors for each FLP assessment were constructed. The next step involved analyzing all the input models by searching all possible predictor combinations and evaluating the informational contribution of each set using the Akaike Information Criterion (AIC). Models with the highest information load and the smallest quantity of predictors (the “best predictor model”) were selected for further analysis.

## 3. Results

Table 1 presents the results of the descriptive statistics (the means and standard deviations) for all studied variables. 

The results of the multiple regression analysis (only the significant best predictor models for FLP assessments) are presented in Table 2. The multiple correlation coefficients between the dependent variable and the predictors for most of the models are significant according to Fisher’s *F*-test (except the models for Reading and Writing). However, most of the determination coefficients (*R*^2^) are low (explaining less than 10% of the variance). Given that FLA success depends on many factors other than creativity and personality traits, these results are satisfactory. 

Table 2 shows that the regression model with the highest determination coefficients (*R*^2^
*=* 0.229) was obtained for the Creativity assessment according to the FLP Scale. This model contains three significantly positive predictors, including two creativity indicators by ATTA (Figural Creativity #1 and Fluency) and one FFM trait (conscientiousness). Consequently, the English teachers’ assessments of their students’ creativity in learning English truly reflect their level of creativity obtained through testing (at least for the result of one task on figural creativity and the overall fluency of their answers). At the same time, the teachers evaluated more conscientious students as more creative, which, in our opinion, is a subjective assessment and does not always correspond to reality.

The regression model for the Total Score according to the FLP Scale explains 15% of the variance of this important variable (*R*^2^
*=* 0.150). Like the previous model, this model contains three significant predictors, including two creativity indicators by ATTA (Figural Creativity#1 has a positive impact and Originality has a negative impact) and one FFM trait (conscientiousness). Thus, teachers more highly rated the total FLP among students with greater expressed figural creativity and conscientiousness, but less expressed originality in solving creative tasks. The regression model for Communication (communicative skills in English) has determination coefficients (*R*^2^
*=* 0.144) that are close to the previous ones and includes only two creativity indicators (according to ATTA) as significant predictors (Figural Creativity #1 has a positive impact and Originality has a negative impact). 

Next, the regression model for Initiative and activity in the classroom explains 13.8% of the variance (*R*^2^
*=* 0.138), and includes three creativity indicators from ATTA as significant predictors *(Figural Creativity #2 and Fluency* have a positive impact; Originality has a negative impact). Finally, one more regression model explains more than 10% of the variance (*R*^2^
*=* 0.138) of the FLP assessment for Final Grades in English. This model contains only two significant predictors: Verbal Creativity, which has a negative impact, and Fluency, which has a positive impact. It is surprising that verbal creativity has a negative impact on academic achievement in English, but this fact, in our opinion, corresponds to the negative impact of originality on the Total Score according to the FLP Scale. The remaining regression models explain less than 10% of the variance of the FLP assessments and include 1–2 significant predictors (more often indicators of creativity than FFM traits).

In accordance with the purpose of the present study, we are primarily focused on comparing the impacts of creativity indicators and personality traits on the FLP assessments. Table 2 shows that one of the most significant predictors for the majority of FLP assessments is Figural Creativity #1 (the result of ATTA’s second activity); it has a significant positive impact on the teachers’ assessments of Audition, Reading, Vocabulary, Communication, Phonetics, Creativity, and Total Score, according to the FLP Scale. Originality is also a significant predictor for several FLP assessments (Audition, Speaking, Communication, Initiative, Diligence, Creativity, and Total Score), but has a negative impact. Fluency has a positive significant impact on Speaking, Initiative, Creativity, and Final Grades. Figural Creativity #2 has a positive significant impact only on Initiative, Verbal Creativity has a negative significant impact only on Final Grades, and Flexibility has a negative significant impact only on Vocabulary. 

In comparison to the indicators of creativity, personality traits have a much smaller impact on FLP assessments. Extraversion is included in the models for Speaking, Communication, Diligence, Creativity, and Total Score, but has a non-significant and negative impact (*p* > 0.05). Agreeableness is included in only two models for Writing and Final Grades, but also has a non-significant and negative impact (*p* > 0.05). Neuroticism, included in two models of personality characteristics, is associated with the FL learning process (Initiative and Creativity in learning English according to the teachers’ evaluation), but has a non-significant and negative impact (*p* > 0.05). Openness, which has a non-significant and negative impact (*p* > 0.05), is included only in one model for Phonetics. Conscientiousness is the only FFM trait that has a significant and positive impact on FLP assessments (Diligence, Creativity, and Total Score).

## 4. Discussion

The purpose of this study is to consider and compare creativity and personality traits’ impacts on the level of FLP among Linguistics students. We assumed that, among personality traits, conscientiousness and openness have a positive effect on FLA success, while creativity indicators have a controversial effect, especially on the teachers’ assessments. On average, this hypothesis has been confirmed: Creativity indicators have a stronger but more contradictory impact on FLP assessments than personality traits. On the one hand, Fluency, and especially Figural Creativity #1, are significant and positive predictors of the level of FLP. On the other hand, Originality is a significant and negative predictor for most FLP assessments, and Verbal Creativity is a significant and negative predictor for final grades in English. The opposite impacts of Fluency and Originality on the level of FLP corresponds to the findings by Albert and Kormos [41]. As expected, for personality traits, only conscientiousness has a significant and positive impact on several FLP indicators, including the Total Score according to the FLP Scale. This fact corresponds to the data of numerous studies showing that conscientiousness is the most universal and positive predictor of academic achievement for most fields of study, including FL/SL, in different national samples [15,16,19]. A joint examination of creativity and personality traits as FLA success predictors allowed us to establish that teachers evaluate more conscientious students as more creative, but, at the same time, conscientiousness and Originality have opposite impacts on FLP assessments.

Thus, our findings show that creativity can have not only a positive, but also a negative impact on FLA success, which contradicts numerous data on the positive relationship between creativity and bilingualism [36,37] and research findings on the positive correlation of creativity and FL/SL proficiency [32,38,39].

In our opinion, these contradictions could have the following explanations:The FLP assessments used in our study are largely based on teachers’ opinions, which may be subjective and may not reflect real English proficiency levels (this is one of the main limitations of our research). In this future, it will be necessary to use more objective assessment methods, such as TOEFL, MLAT, etc.English teachers can ambiguously and even negatively assess the manifestations of students’ creativity during the learning process. In particular, as our results show, teachers better appreciate the achievements of students with higher non-verbal creativity and fluency (generally combined with a high level of conscientiousness), but assign lower marks to students with higher indicators of verbal creativity and originality. This contradiction may be due to the fact that creativity is not sufficiently included in the education system, despite the need for creativity, as noted by many scholars and practitioners [19,34,37]. Indeed, contemporary educational and grading systems are more focused on correct and meticulous task completion than the development of students’ creative thinking, non-typical problem solving, soft skills, and so on. This assumption is consistent with the results of other studies, which show that students’ academic achievements are more closely associated with conscientiousness and persistence, rather than with initiative, openness, curiosity, etc. [15,19,20,21,23,24]. Overcoming these problems is not so much a psychological and educational problem as it is a social one.The Russian educational and teaching system has some special aspects that could have influenced our study. Since such studies have not been carried out before, it is difficult to compare our results with those of other researchers to provide a more comprehensive outlook on cultural features.

Summing up all the findings and limitations of our research, we have determined the following future prospects: (1) Studying FLA predictors among non-linguistics students; (2) expanding the sample and balancing its female-to-male ratio; (3) considering other languages studied as second (or third) foreign languages; (4) using additional measurement methods for both creativity and FLA, as well as other methods for statistical analysis.

## Figures and Tables

**Table 1 behavsci-10-00035-t001:** The Means and Standard Deviations (SD) of all Studied Variables, raw scores.

Techniques	Variables (Abbreviations)	Mean	SD
*Five Factor Inventory*	Neuroticism	35.87	7.49
	Extraversion	40.19	8.06
	Openness	40.11	5.17
	Agreeableness	39.97	5.96
	Conscientiousness	41.87	8.27
*ATTA*	Verbal creativity activity (Verbal Creativity)	3.02	2.24
	Figural creativity activity #1 (Figure Creativity#1)	2.88	2.43
	Figural creativity activity #2 (Figure Creativity#2)	6.05	3.90
	Fluency	10.71	5.00
	Originality	4.75	3.15
	Elaboration	4.27	3.68
	Flexibility	1.97	1.10
	Creativity Index (CI)	31.24	13.01
*FLP Scale*	Speed and accuracy of perception of speech by ear (Audition)	4.03	0.83
	Fluency of reading in English (Reading)	4.29	0.73
	Lexical and grammatical correctness of writing (Writing)	4.05	0.76
	Vocabulary (Vocabulary)	3.95	0.72
	Lexical and grammatical correctness of oral speech (Speaking)	3.91	0.74
	Level of communication skills in English (Communication)	4.02	0.84
	Pronunciation (Phonetics)	3.96	0.82
	Initiative and activity in the classroom (Initiative)	4.14	0.91
	Diligence and care in preparing homework (Diligence)	4.22	0.97
	Creativity while studying English (Creativity)	3.66	1.41
	Total Score (Total Score)	40.23	6.44
Semester final grades by English (Final Grades)		83.24	13.69

**Table 2 behavsci-10-00035-t002:** Significant best predictor regression models for FLP assessments.

Variable	Summary of Model			Coefficients			
	*R* ^2^	*F*	*p*-Value	Estimate	Std. Error	*t*-Value	*p*-Value
Audition	0.094	6.521	0.002 *				
(Intercept)				3.989	0.139	28.676	0.000 *
Figure Creativity #1				0.102	0.030	3.354	0.001 *
Originality				−0.053	0.023	−2.263	0.025 ^*^
Vocabulary	0.087	2.922	0.023 *				
(Intercept)				3.945	0.171	23.034	0.000 *
Figure Creativity #1				0.068	0.026	2.529	0.012 *
Fluency				0.029	0.018	1.635	0.104
Originality				−0.036	0.021	−1.713	0.089
Flexibility				−0.168	0.084	−1.986	0.049 *
Speaking	0.082	2.179	0.060				
(Intercept)				4.398	0.365	12.021	0.000 *
Figure Creativity #2				0.027	0.019	1.451	0.149
Fluency				0.040	0.019	2.082	0.039 *
Originality				−0.047	0.023	−2.046	0.042 *
Flexibility				−0.149	0.091	−1.631	0.105
Extraversion				−0.014	0.008	−1.766	0.079
Communication	0.144	5.167	0.000 *				
(Intercept)				4.365	0.389	11.204	0.000 *
Figure Creativity #1				0.113	0.030	3.768	0.000 *
Fluency				0.020	0.014	1.454	0.148
Originality				−0.058	0.023	−2.476	0.014 *
Extraversion				−0.015	0.008	−1.772	0.078
Phonetics	0.074	3.291	0.022 *				
(Intercept)				4.932	0.559	8.812	0.000 *
Figure Creativity #1				0.066	0.030	2.187	0.030 *
Originality				−0.034	0.023	−1.431	0.154
Openness				−0.024	0.013	−1.794	0.075
Initiative	0.138	3.919	0.002 *				
(Intercept)				4.378	0.409	10.699	0.000 *
Verbal Creativity				−0.172	0.098	−1.751	0.082
Figure Creativity #2				0.056	0.022	2.541	0.012 *
Fluency				0.041	0.015	2.625	0.009 *
Originality				−0.083	0.028	−2.974	0.003 *
Neuroticism				−0.014	0.010	−1.398	0.164
Diligence	0.088	2.969	0.022 *				
(Intercept)				4.211	0.539	7.811	0.000 *
Figure Creativity #2				0.037	0.024	1.562	0.120
Originality				−0.077	0.029	−2.609	0.010 *
Extraversion				−0.019	0.011	−1.729	0.086
Conscientiousness				0.023	0.011	2.014	0.046 *
Creativity	0.229	5.083	0.000 *				
(Intercept)				3.464	1.375	2.518	0.013 *
Figure Creativity #1				0.204	0.0491	4.159	0.000 *
Fluency				0.079	0.033	2.353	0.020 *
Originality				−0.065	0.038	−1.700	0.091
Flexibility				−0.235	0.157	−1.489	0.139
Neuroticism				−0.029	0.019	−1.551	0.123
Extraversion				−0.031	0.016	−1.936	0.055
Conscientiousness				0.044	0.017	2.559	0.011 *
Total Score	0.150	4.309	0.001 *				
(Intercept)				37.710	3.524	10.698	0.000 *
Figure Creativity #1				0.756	0.232	3.261	0.001 *
Fluency				0.180	0.109	1.651	0.101
Originality				−0.542	0.179	−3.021	0.003 *
Extraversion				−0.140	0.074	−1.899	0.059
Conscientiousness				0.158	0.072	2.199	0.029 *
Final Grades	0.107	2.934	0.015 *				
(Intercept)				91.726	0.987	9.791	0.000 *
Verbal Creativity				−3.356	0.151	−2.230	0.027 *
Figure Creativity #2				0.570	0.340	1.675	0.096
Fluency				0.491	0.240	2.045	0.043 *
Originality				−0.669	0.434	−1.541	0.125
Agreeableness				−0.299	0.202	−1.477	0.142

* *p* < 0.05.

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
