# Peer review of "Creativity and Personality Traits as Foreign Language Acquisition Predictors in University Linguistics Students"

_behavsci, 2020, doi:10.3390/bs10010035_

Round 1

Reviewer 1 Report

It needs to be clear whether the paper is looking at SLA or FLA. The paper talks about these two concepts as if they are the same thing (for example in the opening line of the Abstract and the Introduction, but then in other places (such as line 35, 90, 224, 228), only SLA is referred to. The authors need to be clear about whether they are referring to SLA or FLA, and make the necessary revisions in the paper, based on how they define each concept.

The comment  "authors' expert scale" is subjective and should be rewritten. In addition, the use of the word 'expert" throughout the paper is problematic.

Line 17: "to assess SLA..." is problematic. Surely it is the level of language proficiency that is being assessed, and not the level of acquisition??

Line 15 and 47 and 52 and 57, 76, 213: Are they students of linguistics or language students? A linguistics student need not have any language other than a mother tongue. Are they studying languages or are they studying linguistics?

Line 35-36 : It seems odd to list "language abilities" as an indicator of SLA success, unless you are referring to the extent of a student's linguistic repertoire, which of course can be an indicator of SLA success.

Can the authors check the reference for McCrae and Costa [8] [9], as  I cannot find reference to Costa.

FFM needs to be explained before it is used as an acronym 

Line 71 and 228: Is the author saying that bilingualism is the same as mutlilingualism?

Line 14 and 88: give full  breakdown of numbers (82% female, 18% male).

Although there is a brief comment at the end of the paper on trying to get a better gender balance for future studies, a brief comment in the discussion on whether females are more creative than males would be interesting, and consequently, on what impact this would has on the findings of the current study.

Use of English is problematic at times, and needs to be proofread by a native English speaker:

Line 15 and Line 16: these lines contain no main verb in the sentence. In both cases, the word "IS" should be inserted before "measured".

Line 38: commas are needed for relative clauses. The phrase "on the one hand" line 38 and "on the other hand" line 40, both need to be followed by a comma.

Line 42: remove "the" before SLA

Line 46: "that consciousness impacts on..."

Line 45: "researches" should be "research"

Line 48: remove "the" after "final"

Line 60: needs to be rewritten....factors appears in the plural, but only one factor is referred to. In addition"can contradictory predict" does not make sense

Line 65: insert "the" before "implementation"

Line 77: "as well as" needs to be replaced by "or"

Line 82: why bold and italics?

Line 82: remove "the" before creativity

Line 85: "can be controversial"

Line 94: replace "also" with "in addition"

Line 101: This sentence does not make sense in English

Line 103: using a 5 point Likert scale

Line 110: the first test calls for....

Line 112: comma needed after "task"

Line 114: tests call for

Line 114: figural reponse???

Line 116: requires the respondents to name these pictures

Line 116 / 121: measurements (not measures)

Section 2.2: why use italics throughout the paragraph

Line 135: This data....

Line 148: remove "an"

Line 168-170: This sentence has no main clause.

Line 174-176: This sentence is not clear in English

Line 190: Focus ON...(not in)

Line 202 and 204: included in

Line 205: " included in"

Line 230: ARE largely based...

Line 231: and DO not reflect....

Line 235: they appreciate more the...

Line 326 " but they lower assessed" does not make sense in English

Author Response

Dear Reviewer,

Thank you for your critical reading but friendly and constructive suggestions.

Please find our revised manuscript in the attached file. We would like to pay your attention to the colours of corrections.  All our corrections were made in red colour and blue ones were made by the native English speaking editor by "Track Changes" function.

Please find our detailed replies in the attached file.

Reviewer 2 Report

The topic of the manuscript addresses the examination of personality traits and creativity as predictors of success in second language acquisition (SLA). Taken into consideration that the search for predictors of success in language learning is an important problem, it definitely deserves greater attention from linguists and language experts. Thus, the study is potentially very valuable.

Nevertheless, there are several points which I believe need some improvement. The theoretical background is rather pertinent but I missed the justification of the research as it was not very clear for me why personal traits and creativity are studied together. Probably, it could be a good idea to explain the relation between these aspects and clarify if there were studied together (if yes, explain the results). The aims of the research should be better defined and go in line with previous studies in this area. I would appreciate the author to describe in a better way in which way this study broadens our knowledge of the subject.

In respect of the methodology, the tools of the research and the procedures are rather clearly described. There is also a systematic analysis of the data collected by the authors, including the Abbreviated Torrance Test for Adults (ATTA) for measuring creativity and the Five-Factor Inventory (FFI) for measuring personality traits. One thing that was not very clear for me is the choice of language assessment criteria. Why these criteria have been taken into consideration?

The discussion section looks more like a Conclusion as it provides some concluding remarks.

Finally, please, take into consideration that the manuscript requires extensive language proofreading.

Author Response

(The authors gave the same response as above.)
